



# Evaluating consistency between total column $CO_2$ retrievals from OCO-2 and the *in-situ* network over North America: Implications for Carbon flux estimation

Bharat Rastogi[1,2], John B Miller[2], Micheal Trudeau[1,2], Arlyn E Andrews[2], Lei Hu[1,2], Marikate Mountain[3], Thomas Nehrkorn[3], John Mund[2], Kaiyu Guan[4], and Caroline B Alden[1,2]

[1]Cooperative Institute for Research in Environmental Sciences (CIRES), University of Colorado Boulder, CO, 80309
[2]Global Monitoring Laboratory, National Oceanic and Atmospheric Administration, Boulder, CO 80305
[3]Atmospheric and Environmental Research, Lexington, MA 02041
[4]Department of Natural Resources and Environmental Sciences, College of Agriculture, Consumer, and Environmental Sciences, University of Illinois at Urbana-Champaign, Urbana, IL 60801

**Correspondence:** Bharat Rastogi (bharat.rastogi@noaa.gov)

**Abstract.** Feedbacks between the climate system and the carbon cycle represent a key source of uncertainty in model projections of Earth's climate, in part due to our inability to directly measure large scale biosphere-atmosphere carbon fluxes. *In-situ* measurements of $CO_2$ mole fraction from surface flasks, towers and aircraft are used in inverse models to infer fluxes, but mea-

surement networks remain sparse, with limited or no coverage over large parts of the planet. Satellite retrievals of total column $CO_2$ ($X_{CO_2}$), such as those from NASA's Orbiting Carbon Observatory-2 (OCO-2), can potentially provide unprecedented global information about $CO_2$ spatiotemporal variability. However, for use in inverse modeling, data need to be extremely stable, highly precise and unbiased to distinguish abundance changes emanating from surface fluxes from those associated with variability in weather. Systematic errors in $X_{CO_2}$ have been identified and, while bias correction algorithms are applied

globally, inconsistencies persist at regional and smaller scales that may complicate or confound flux estimation. To evaluate $X_{CO_2}$ retrievals and assess potential biases, we compare OCO-2 *v10* retrievals with *in-situ* data-constrained $X_{CO_2}$ simulations over North America estimated using surface fluxes and boundary conditions optimized with observations that are rigorously calibrated relative to the WMO X2007 $CO_2$ scale. Systematic errors in simulated atmospheric transport are independently evaluated using unassimilated aircraft and AirCore profiles. We find that the global OCO-2 *v10* bias correction shifts the distri-

bution of retrievals closer to the simulated $X_{CO_2}$, as intended. Comparisons between bias corrected and simulated $X_{CO_2}$ reveal differences that vary seasonally. Importantly, the difference between simulations and retrievals is of the same magnitude as the imprint of recent surface flux in the total column. This work demonstrates that systematic errors in OCO-2 *v10* retrievals of $X_{CO_2}$ over land can be large enough to confound reliable surface flux estimation and that further improvements in retrieval and bias correction techniques are essential. Finally, we show that independent observations, especially vertical profile data, such

as from NOAA's aircraft and AirCore programs are critical for evaluating errors in both satellite retrievals and carbon-cycle models.





# 1 Introduction

Interannual variability in the growth rate of atmospheric $CO_2$ is largely driven by variability in uptake and release by terrestrial
ecosystems (Heimann and Reichstein, 2008; Piao et al., 2020). Oceanic fluxes also respond to variability in climate (e.g., DeVries et al., 2019; Riebesell et al., 2009), but the amplitude of oceanic flux variability is thought to be considerably less than for terrestrial fluxes. While individual component fluxes (e.g., photosynthesis) are currently not directly measurable at scales larger than leaf or soil chambers, well-calibrated and precise measurements of $CO_2$ have allowed us to track the accumulation of this greenhouse gas in the atmosphere and associated radiative feedbacks on the global climate, as well as its spatiotemporal variability (e.g., Tans et al., 1989). These measurements continue to provide valuable insights into surface flux processes and feedbacks (e.g., Tans et al., 1990; Ballantyne et al., 2012; Keeling et al., 2017; Arora et al., 2020). Observed spatial and temporal gradients in $CO_2$ mole fraction (relative to dry air) can be combined with a numerical model of atmospheric transport to infer surface fluxes (i.e. exchange of $CO_2$ between the atmosphere and the underlying ocean or land surface), in a "top-down" or inverse modelling framework (e.g., Peters et al., 2007; Gurney et al., 2002). An ever-increasing global greenhouse gas measurement network and progress in modelling techniques have tremendously improved our understanding of surface processes. However, measurement networks remain sparse and continue to under-sample large parts of the world, including large parts of North America, which can limit our understanding of surface flux processes in those regions. Furthermore, incompatibility across datasets that arises from inconsistent calibrations or systematic errors can significantly corrupt surface flux estimates – leading to inaccurate models of carbon-climate interactions and subsequent errors in climate forecasts.

Satellite retrievals of total column $CO_2$ mole fraction ($X_{CO_2}$), such as those from NASA's Orbiting Carbon Observatory-2 (OCO-2), have the potential to provide unprecedented information about spatio-temporal patterns and variability in the Earth's atmosphere. However, space-based observations of $X_{CO_2}$ must be extremely stable, highly precise and free from bias to detect and quantify abundance changes caused by a change in surface fluxes (Rayner and O'Brien, 2001; Olsen, 2004; Miller et al., 2007; Houweling et al., 2003). This is critical, as even large surface flux signals are substantially diluted in the total column and largely obscured by meteorological variability (Basu et al., 2018; Feng et al., 2019).

*In-situ* measurements that comprise global networks, such as NOAA's Global Greenhouse Reference Network (https://www.esrl.noaa.gov/gmd/ccgg/ggrn.php), are rigorously evaluated and carefully calibrated relative to the World Meteorological Organization (WMO) calibration scale (data used here are reported on the X2007 scale), thus ensuring the fidelity of these measurements over timescales of seasons to decades (Andrews et al., 2014; Hall et al., 2020). The "open-path" nature of space-based $X_{CO_2}$ measurements however, does not allow for direct calibration. Satellite retrievals of $X_{CO_2}$ require complicated models of atmospheric radiation and are sensitive to a host of assumptions about aerosols, clouds, interference of jointly retrieved parameters, surface properties and details of the instrumentation (Kulawik et al., 2019). Moreover, sensors typically degrade over time, and limited information is available to characterize resulting time-dependent systematic errors. Post-launch



data corrections are performed if and when biases (O'Dell et al., 2018) and errors (Kiel et al., 2019) are identified, but are severely limited due to the sparsity of calibrated *in-situ* vertical profile observations.

Currently, satellite derived $X_{CO_2}$ retrievals are linked to the WMO scale most directly through a limited set of *in-situ* profiles obtained over a network of ground-based Fourier Transform Infrared Spectrometers that comprise the Total Carbon Column Observation Network (TCCON; Wunch et al., 2017). However, TCCON itself provides remotely sensed information about $X_{CO_2}$, and TCCON retrievals undergo a complex validation and bias correction routine (Wunch et al., 2010) that links these retrievals to the WMO scale. Moreover, TCCON sites are few. Issues with validation of OCO-2 retrievals via TCCON have been identified- seasonal and site-dependent biases have been reported (Wunch et al., 2017, 2015), raising questions about the adequacy of this network to validate satellite derived $X_{CO_2}$ products (Basu et al., 2013). This is especially important since systematic bias corrections of $X_{CO_2}$ from TCCON data are developed over small spatio-temporal scales and extrapolated globally (Wunch et al., 2011) for retrievals over land and ocean respectively (O'Dell et al., 2018). OCO-2 retrievals are additionally corrected for bias by comparing with 4-D $CO_2$ mole fraction fields from global inverse models, and a small-area approximation, but both methods are prone to smoothing across fine-scale variability in $X_{CO_2}$ (O'Dell et al., 2018; Corbin et al., 2008). While bias correction generally reduces inferred surface flux uncertainty when retrievals are assimilated in atmospheric inversions (Basu et al., 2013), even small retrieval errors can lead to large errors in inferred flux (Takagi et al., 2014; Chevallier et al., 2014). Biases in $X_{CO_2}$ from OCO-2, hereafter $X_{CO_2}^{ret}$, have been identified, and found to be related to surface (e.g. pressure, albedo) and atmospheric (e.g. aerosol loading, sky condition) properties (Kiel et al., 2019). However, systematic biases not accounted for in the *v10* bias correction approach persist and therefore the measurement uncertainty associated with individual retrievals is believed to be at least twice the value currently reported in the OCO-2 data files (Eldering et al., 2017).

Thus, a dynamic method to routinely evaluate satellite retrievals is necessary. In this study, we propose such a method that takes advantage of the relatively dense *in-situ* network of $CO_2$ mole fraction measurements over North America, and leverage an ensemble of optimized flux estimates derived using the high resolution CarbonTracker-Lagrange $CO_2$ inverse modeling framework (Hu et al., 2019). We demonstrate an approach for constructing $X_{CO_2}$ that offers optimal consistency with *in-situ* measurements of $CO_2$ dry air mole fraction calibrated relative to the WMO X2007 scale (Hall et al., 2020). We compare our simulated $X_{CO_2}$ retrievals with the *v10* OCO-2 product, evaluate the extent to which observed differences are consistent with rigorous uncertainties on the simulated fields, and potentially correct biases in satellite retrievals. We essentially use the CarbonTracker-Lagrange modeling framework to interpolate the existing *in-situ* measurements to the time and location of OCO-2 retrievals. The *in-situ* measurement uncertainty is generally $\sim 0.15$ [ppm] (Andrews et al., 2014), and we have carefully accounted for uncertainty in regional boundary conditions and uncertainties in the optimized fluxes. While a single realization of the simulated atmospheric transport is used here (a limitation of the approach that we aim to address in future work by using transport ensembles), we evaluate transport uncertainty using independent vertical *in-situ* profiles of $CO_2$. CarbonTracker-Lagrange simulations are driven by meteorological simulations from the Weather Research Forecast model system optimized for particle dispersion modeling (Nehrkorn et al., 2010) with resolution of 10 [km] over continental U.S., and 30 [km] over the rest of North America, considerably higher than global *in-situ* informed simulations that have so far been used for OCO-2 evaluation (Kiel et al., 2019; Crowell et al., 2019; Miller and Michalak, 2020). Simulated $X_{CO_2}$ are





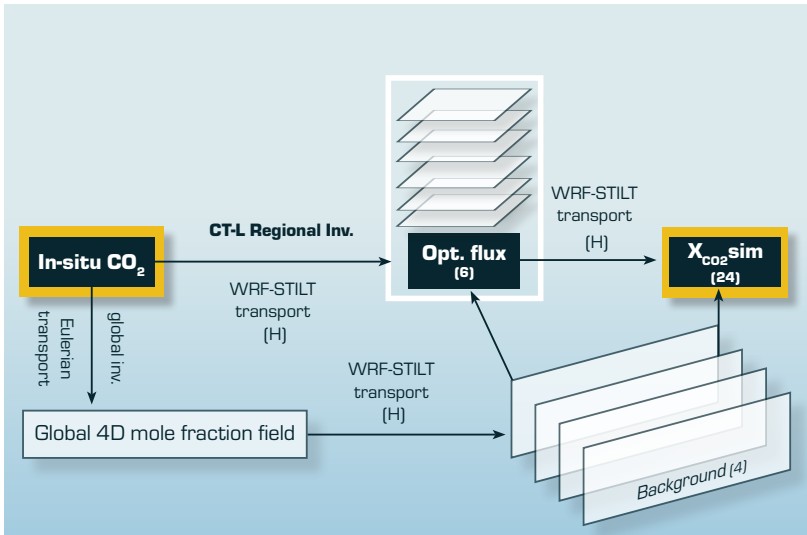

**Figure 1.** Flowchart linking *in-situ* measurements of $CO_2$ to simulated columns.

compared with OCO-2 *v10* retrievals, both before and after global bias correction, thus providing an independent evaluation of

90 the global bias correction over North America. In principle, differences can result from errors in the retrievals or from errors in the CarbonTracker-Lagrange modeling framework, but we show that for certain seasons these differences are unlikely to result from the latter. North America is a useful test-bed for evaluating consistencies and for developing improved model simulations and retrieval bias correction strategies, given the relatively dense sampling network (compared to other regions) and that the best surface flux estimates are likely to come from approaches that combine *in-situ* measurements and satellite retrievals (Basu

95 et al., 2013; Fischer et al., 2017; Byrne et al., 2020). The US Inter-Agency North American Carbon Program (Wofsy et al., 2002) has supported intensely focused research for nearly two decades and has resulted in a wealth of datasets and model:data fusion activities that have informed model development.

## 2 Methods

Simulated $X_{CO_2}$, hereafter "$X_{CO_2}^{sim}$", is constructed by estimating impact of different surface fluxes ($\Delta_{CO_2}^{flux}$) on the total column.

100 This involves imposing a time, latitude, longitude, and altitude dependent lateral boundary condition or background, which accounts for changes in $X_{CO_2}^{sim}$ originating outside our model domain. The chain of events that link the *in-situ* data with the simulations is shown in fig. 1. We use multiple ensembles of surface flux and boundary conditions to assess uncertainty in each. Comparisons with independent unassimilated aircraft and AirCore profiles are used to assess combined random and systematic errors in surface flux, background and atmospheric transport.





## 2.1 Convolution Method

We follow the recommended protocol for comparing satellite retrievals with modeled $CO_2$ columns from the Atmospheric $CO_2$ Observations from Space (ACOS) retrieval algorithm (O'Dell et al., 2012).

$$X_{CO_2}^{sim} = \sum_{i=1}^{N} w_i \left[ a_i.\chi_{CO_2,i}^{model} + (1 - a_i).\chi_{CO_2,i}^{pri} \right] \tag{1}$$

Here, $X_{CO_2}^{sim}$ [$ppm$], the total column $CO_2$, is computed as a *pressure weighted* sum of the modeled column ($X_{CO_2}^{model} = \sum_{i=1}^{N} w_i.\chi_{CO_2,i}^{model}$), comprising $N$ model (i.e., not OCO-2) levels from the surface to the top of the atmosphere ($0.01$ [$hPa$]). $\chi_{CO_2}^{model}$ is convolved with the OCO-2 averaging kernel profile ($a_i$) and the OCO-2 prior profile ($\chi_{CO_2}^{pri}$) and summed according to a pressure weighting function ($w$; identical to $h$ in Connor et al., 2008). $w$ is calculated as:

$$w = \sum_{i=1}^{N} w_i = \sum_{i=1}^{N} \left| \left( -p_i + \frac{p_{i+1} - p_i}{ln(\frac{p_{i+1}}{p_i})} \right) + \left( -p_i + \frac{p_i - p_{i-1}}{ln(\frac{p_i}{p_{i-1}})} \right) \right| \frac{1}{p_{surf}}, \tag{2}$$

where $p_i$ and $p_{surf}$ are WRF modeled pressure at level $i$ and at the surface respectively. The profile sum of $w$ is always unity.

$\chi_{CO_2,i}^{model}$ [$ppm$] is constructed as:

$$X_{CO_2}^{model} = \sum_{i=1}^{N} \chi_{CO_2,i}^{bkg} + \sum_{i=1}^{N-3} \Delta_{CO_2,i}^{flux} \tag{3}$$

Here, $\chi_{CO_2,i}^{bkg}$ [$ppm$] represents background (i.e., lateral boundary condition, described in sec. 2.4) and $\Delta_{CO_2,i}^{flux}$ [$ppm$] denotes the impact of surface flux at level $i$ of the column. $\Delta_{CO_2,i}^{flux}$ is computed at discrete levels from the surface to 14 [$km$], whereas $\chi_{CO_2,i}^{bkg}$ is computed at 3 additional levels. These additional levels represent the upper troposphere and the stratosphere, where influence of recent surface flux is assumed to be zero. $\Delta_{CO_2,i}^{flux}$ is estimated as:

$$\Delta_{CO_2,i}^{flux} = H_i(s_{bio} + s_{ff} + s_{bmb} + s_{ocn}), \tag{4}$$

where $H_i$ [$ppm\ CO_2.(\mu\ mol\ m^{-2}\ s^{-1})^{-1}$] represents the sensitivity at pressure level $i$ of simulated $X_{CO_2}$ to upwind surface fluxes (detailed in sec. 2.3), and s indicates surface flux [$\mu\ mol\ m^{-2}\ s^{-1}$]. $s_{bio}$ denotes net ecosystem exchange (i.e. the sum of ecosystem photosynthesis and respiration), $s_{bmb}$ denotes biomass burning, $s_{ff}$ corresponds to fossil fuel emissions and $s_{ocn}$ is the net ocean-atmosphere flux. Fluxes are described in detail in sec. 2.5.

## 2.2 OCO-2 retrievals and receptor selection criterion

We construct $X_{CO_2}$ for valid *land-nadir* and *land-glint* $X_{CO_2}^{ret}$ from the *v10* data product (Osterman et al., 2020) over North America between September 2014 and August 2015. Globally, OCO-2 retrievals are primarily obtained in three operation





modes: Land Glint, Land Nadir, and Ocean Glint. Additionally, retrievals are obtained in "target" mode, for evaluation against TCCON, and a "transition" mode, where the sensor switches between modes. While the "nadir" mode is mostly used over land, over darker ocean the satellite sensor is able to receive a higher fraction of directly reflected sunlight in a separate "glint" mode (Eldering et al., 2017). Here, we evaluate soundings over land obtained in both *nadir* and *glint* modes. Additionally, only retrievals that passed the aerosol and cloud-screening filers are considered (i.e., quality flagged as 0, or "good").

Over North America, there can be a few thousand to tens of thousands of valid retrievals on any given day (Fig. 2a). Each retrieval covers an area of approximately $1.29\,[km] \times 2.25\,[km]$ on the surface. Individual satellite retrievals are known as *footprints*. The satellite collects 8 simultaneous *footprints*, and the next row of *footprints* are spaced $300\,[ms]$ apart. For each satellite overpass (*along-track*), we select locations every $2\,[s]$ over the continental U.S. (i.e.,$\sim 12\,[km]$) and $4\,s$ (i.e.,$\sim$ $24\,[km]$) over the rest of North America (Fig. 2b). These locations are usually called receptors in a Lagrangian particle dis-

persion model (LPDM) as they represent locations and times from which a set of particles are released and tracked back in time. In an LPDM, an ensemble of particles is released from each receptor, and the residence time of particles in the planetary boundary layer is used to calculate sensitivities describing the relationship between upwind surface fluxes and mole fraction at the receptor location. Ultimately, a library of sensitivity arrays is generated corresponding to $X_{CO_2}$ retrieval locations. Note that these sensitivity arrays are sometimes called footprints or influence functions. Here we avoid this use of footprints so as

not to create confusion with OCO-2 footprints (i.e. scenes). This method enables improved simulation of *near-field* transport compared to Eulerian gridded models, as particle locations are not restricted to grid-boxes and meteorological fields can be interpolated to sub-grid scale locations (Lin et al., 2003), and has been used extensively in estimating regional trace gas fluxes in inverse models using *in-situ* measurements (e.g., Schuh et al., 2009; Gourdji et al., 2012; Lauvaux et al., 2013; Alden et al., 2016; Hu et al., 2019).

In this study, a vertical profile of receptors corresponding to a range of altitudes is created corresponding to the location of a single valid satellite retrieval. We preferentially select receptors that are in the middle of the OCO-2 *track* to provide the most spatially representative sample and minimize footprint dependant biases (O'Dell et al., 2012). For this analysis, we assume that $X_{CO_2}^{sim}$ from a given receptor location is representative of all $X_{CO_2}^{ret}$ within $1\,[s]$. $\sim 32,000$ unique receptor profiles associated with valid retrievals are created between September 2014 and August 2015 representing $\sim 1.61$ million retrievals that passed

quality flags between September 2014 and August 2015. At each receptor, 24 unique $X_{CO_2}^{sim}$ are created, from combinations of six flux and four background ensemble members (Fig. 1). Using an ensemble of 24 flux-background combinations provides an estimate of unresolved variability in the simulations. In this analysis we report $X_{CO_2}^{sim}$ as the the mean and standard deviation of these simulations. Similarly, $X_{CO_2}^{ret}$ represents the mean of all OCO-2 footprints within $\pm 1\,[s]$ for a selected retrieval.

A profile of receptors for each selected satellite scene consists of discrete altitude levels approximately representing the

lowermost $850\,[hPa]$ of the atmosphere. Models sampled for background estimation are sampled at receptor locations to account for the rest of the column (detailed in sec. 2.4). A similar method (the application of a regional Lagrangian model to estimate source-receptor relationships) was developed recently by Wu et al. (2018). In that study a set of model particles distributed throughout an entire column of air (weighted to appropriately represent the retrieved total column, i.e., "the column receptor") was transported and tracked backward in time and a single surface flux sensitivity array was computed for the total

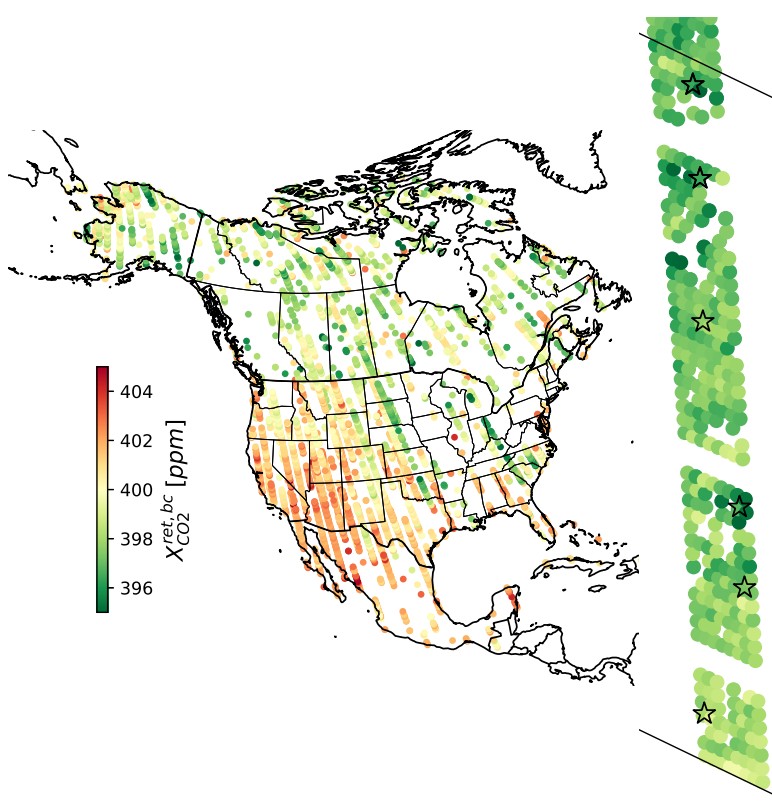

**Figure 2.** All valid satellite retrievals for June 2015 (left). A zoomed-in map (right) shows all retrievals for a section of an individual sounding and locations of *along-track* receptors (stars). Black lines in the zoomed map indicate 1° latitude bands.

column. In contrast, here we establish source-receptor relationships at discrete altitude intervals and then apply appropriate vertical weighting for the column. This method has a higher computational cost and results in larger output files but adds flexibility in the simulation, for instance, a quick recalculation can be performed if the OCO-2 averaging kernel is modified and more importantly, our simulated profiles can be used with experimental retrieval products such as partial column retrievals (e.g., Kulawik et al., 2017).

**2.3    Transport model and constructing column weighted surface-sensitivity arrays: H**

We use output from the Stochastic Time-Inverted Lagrangian Transport (STILT) particle dispersion model (Lin et al., 2003) driven with high resolution meteorological fields from a customized implementation of the Weather Research and Forecasting (WRF) model (Nehrkorn et al., 2010). We use the WRF-STILT modeling configuration developed for the CarbonTracker-Lagrange modeling framework (Hu et al., 2019).





175 The high spatial resolution of regional models provides an appropriate framework to investigate relatively fine-scale structure in atmospheric transport (e.g. mesoscale) and atmospheric signatures of surface flux heterogeneity. WRF model fields are computed at $10\,[km]$ spatial resolution over temperate North America and $30\,[km]$ spatial resolution outside of temperate North America. STILT is run off-line (i.e. driven by archived hourly WRF output) and trajectories are computed backwards in time from each receptor location. STILT surface-sensitivity arrays represent simulated upwind surface flux influence for 10

180 days prior to each observation at $1° \times 1°$ spatial and 1 hourly temporal resolution. A library of WRF-STILT surface-sensitivity arrays is pre-computed, archived, and can be efficiently convolved with independently estimated surface fluxes. This is in contrast to most modern Eulerian model $CO_2$ simulations, where the transport model needs to be rerun whenever a new surface flux product becomes available. STILT surface-sensitivity arrays have units of $[ppm(\mu\,mol\,m^{-2}\,s^{-1})^{-1}]$.

 The OCO-2 retrieval is a *pressure-weighted* mean of $\chi_{CO_2}$ obtained at 20 equally spaced pressure boundaries from the

185 surface to the top of the atmosphere. STILT receptors however, are specified not on the pressure grid used for OCO-2 retrievals, but at fixed altitudes above ground level and with high vertical resolution where strong gradients in $CO_2$ are expected (e.g. near the surface, or at the top of planetary boundary layer). Additionally, $X_{CO_2}^{ret}$ is computed from a combination of the signal received by the spectrometer and an *a priori* profile ($\chi_{CO_2}^{pri}$). This approach constrains the uppermost portion of the atmospheric column where the satellite sensor lacks sensitivity. The relative weights of the received signal and $\chi_{CO_2}^{pri}$ are described by the

190 column averaging kernel ($a$; O'Dell et al., 2012), which is computed during the retrieval and archived along with $X_{CO_2}^{ret}$ and $\chi_{CO_2}^{pri}$. Thus, the first step in creating column-weighted surface-sensitivity arrays ($H$) is to interpolate $\chi_{CO_2}^{pri}$ and $a$ onto the STILT grid. Then, a pressure weighting function (eq. 2) is applied to appropriately weight the surface sensitivity obtained from all receptors for each column. The upper $150\,[hPa]$ of the atmosphere is considered as part of the lateral boundary condition (the column weighted background; section 2.4), as sensitivity to recent surface flux is assumed to be zero at these pressure

195 levels and the WRF-STILT framework has not been optimized for upper atmospheric simulations.

## 2.4 Column weighted background: $X_{CO_2}^{bkg}$

The background or lateral boundary condition is an essential component of regional models, required to isolate changes in $CO_2$ from surface fluxes within the model domain. Boundary values need to represent synoptic variability and contributions from surface fluxes outside the model domain and may contribute significantly to uncertainty in modeled $X_{CO_2}$ (Feng et al., 2019).

200 Here, we combine WRF-STILT back-trajectories with 4-D global mole fraction fields from simulations that were optimized using global *in-situ* measurements. From each receptor, 500 back-trajectories (simulating air parcels) are released and tracked backwards in time until the point at which they exit the WRF domain or for the duration of the simulation (10 days). At that coordinate (longitude, latitude, altitude, time) a global 4-D mole fraction field is sampled and that value is considered as the background value for that particle. Background values for all 500 particles are then averaged to calculate the estimated

205 background value for a given receptor. For background estimation, the WRF domain is subset to only include continental North America plus margins along the coast. These strategies minimize transport-related errors in the trajectories that inflate with increasing distance from the receptor and time of release. The background values for each receptor in a column are summed according to the pressure weighting function (eq. 2).





To assess errors due in global 4-D mole fraction fields used to estimate the background, we sample four different models
that are informed by *in-situ* measurements and routinely updated. These include two versions of CarbonTracker (CT2016 and
CT2019B; Peters et al., 2007; Jacobson et al., 2020) from NOAA's Global Monitoring Laboratory (GML), the Copernicus
Atmosphere Monitoring Service reanalysis (CAMS; Chevallier et al., 2019) produced by the European Centre for Medium-
Range Weather Forecasts (ECMWF), and the Jena CarboScope model from the Max Planck Institüt for Biogeochemistry
(Rödenbeck et al., 2020). We evaluate each model against all designated assimilable $CO_2$ data from NOAA GML's GlobalView
Plus version 6.0 data product (Table 1 Schuldt et al., 2020). Assimilable data include assimilated as well as withheld data.
Withheld data are qualitatively equivalent to assimilated data (i.e., they pass the same quality flags) but are excluded and used
to evaluate model results in CarbonTracker (A.R. Jacobson, pers. comm.). Here, over 150,000 ground, tall-tower and aircraft
*in-situ* observations spanning 2014-15 over North America and the eastern North Pacific Ocean are used. Comparisons with
these observations are provided for all assimilable observations, and assimilable observations between $4 - 8\,[km]$.

**Table 1.** Global $CO_2$ 4D mole fraction fields used in this study. Comparisons with GML's GlobalView Plus version 6.0 data product are also
presented. These comparisons are provided for all assimilable observations, and all assimilable observations between $4 - 8\,[km]$ .

| Model | Version | Resolution | Comparison with GV+ 6.0 obs [ppm] | |
|---|---|---|---|---|
| | | | all altitudes | 4 to 8 $[km]$ |
| CarbonTracker | CT2016 | $3° \times 2° \times 3\,hrly$ | 0.49 $\pm 0.02$ | 0.02 $\pm 0.03$ |
| CarbonTracker | CT2019B | $3° \times 2° \times 3\,hrly$ | 0.30 $\pm 0.01$ | 0.05 $\pm 0.03$ |
| CAMS | v18r3 (2019) | $1.9° \times 3.75° \times 3\,hrly$ | -0.73 $\pm 0.01$ | -0.18 $\pm 0.03$ |
| CarboScope | v4.3 (2019) | $6° \times 4° \times 6\,hrly$ | 0.60 $\pm 0.02$ | 0.06 $\pm 0.03$ |

## 2.5 Surface fluxes of $CO_2$: s

We sample optimized and imposed flux fields from regional and global inverse models. Net non-fire terrestrial ecosystem
exchange (i.e., $CO_2$ fluxes from photosynthesis and respiration from autotrophic and heterotrophic sources or $s_{bio}$) are from
NOAA's CarbonTracker-Lagrange (Hu et al., 2019). Briefly, CarbonTracker-Lagrange is a regional atmospheric inverse model
in which biospheric fluxes for North America are optimized using surface-sensitivity arrays from high resolution WRF-STILT
simulations and North American measurements of $CO_2$ from GlobalView+ v2.1 (Cooperative Global Atmospheric Data Inte-
gration Project, 2016), which is composed largely of data from NOAA's Global Greenhouse Gas Reference Network and from
Environment and Climate Change Canada. Observations include flask-air measurements from near-surface and aircraft and
quasi-continuous *in-situ* measurements primarily made on towers. The inversions were run with three different prior estimates
of $s_{bio}$. These included two versions of the Carnegie-Ames Stanford Approach (CASA; Potter et al., 1993) biogeochemical
model runs (CASA GFED-CMS and CASA GFEDv4.1) and the Combined Simple Biosphere/Carnegie-Ames-Stanford Ap-
proach terrestrial carbon cycle model (SiBCASA; Schaefer et al., 2008). Prior error covariance parameters were derived from





maximum likelihood estimation (MLE) with fixed correlation scales of $1000\,[km]$ and $7\,[days]$ and also for optimized correlation scales, for each model runs, resulting in six different posterior estimates of $s_{bio}$. Non-biospheric fluxes include imposed biomass burning ($s_{bmb}$) and fossil fuel ($s_{ff}$) fluxes, and optimized ocean fluxes ($s_{ocn}$) from CT2016. We use the mean of two

fossil fuel emission products ("Miller" and "ODIAC" datasets) and fire emission products ("GFED4.1s" and "GFED-CMS") used in CT2016. All fluxes are $1° \times 1°$ spatial and 3 hourly temporal resolution.

## 3   Results and discussion

### 3.1   Comparing simulated and satellite retrievals

For all retrievals selected over the spatiotemporal domain of this study, the impact of the OCO-2 bias correction is $2.01 \pm$

$0.87\,[ppm]$, and the mean difference between seasons is $0.5\,[ppm]$ (1.76 in winter and spring, and $2.31\,[ppm]$ in summer; blue distributions in Fig. 3). Across seasons, the difference in residuals between $X_{\mathrm{CO_2}}^{ret,bc}$ and $X_{\mathrm{CO_2}}^{sim}$ is significantly lower than that between $X_{\mathrm{CO_2}}^{sim}$ and $X_{\mathrm{CO_2}}^{ret}$: $\mu_{sim-ret} = 2.23 \pm 1.36\,[ppm]$, whereas $\mu_{ret,bc-sim} = -0.22 \pm 1.91\,[ppm]$, i.e., the OCO-2 bias correction brings the distribution of OCO-2 $X_{\mathrm{CO_2}}$ substantially closer to the *in-situ* data-constrained synthetic columns ($X_{\mathrm{CO_2}}^{sim}$), as expected. Residuals are lowest in the Northern Hemisphere summer months of June, July and August

($\mu_{ret,bc-sim} = 0.2 \pm 1.36\,[ppm]$; Fig. 3d) and highest in the winter and spring ($\mu_{ret,bc-sim} = -0.61\,[ppm]$; Fig. 3a). Apart from the summer, mean $X_{\mathrm{CO_2}}^{ret,bc}$ over North America is lower than $X_{\mathrm{CO_2}}^{sim}$.

### 3.2   Spatial patterns

To examine spatial patterns of simulated and retrieved soundings we first sort retrievals in $2° \times 2°$ bins and then average all retrievals (and simulations) in each bin for each season. The spatial extent of OCO-2 soundings varies seasonally (Fig. 4a,e,i,m).

The northern extent of valid retrievals follows the solar declination, as the OCO-2 spectrometer is unable to retrieve a signal over land that is dark or blanketed by snow. Consequently, between September and March, there are few soundings north of the U.S.- Canada border ($\sim 49°$N). Conversely, all of North America is observable between March and August.

Across seasons, $X_{\mathrm{CO_2}}^{sim}$ and $X_{\mathrm{CO_2}}^{ret,bc}$ exhibit broadly similar spatial patterns (Fig. 4a-n). However, residuals between the two reveal that $X_{\mathrm{CO_2}}^{ret,bc}$ is usually lower than $X_{\mathrm{CO_2}}^{sim}$, except in the summer, when a majority of retrievals in the eastern half of the

continent (right of the dotted lines in Fig. 4o) are higher than the simulations. Such coherent spatial differences are not evident when examining the mean bias over the continent (Fig. 3). Importantly, the magnitude of this difference is similar to that of the recent flux signals in the total column (Fig. 4d, h, l, p). For e.g., the mean residuals in the northeast quadrant during the summer (Fig. 4o) are $0.79 \pm 1.63\,[ppm]$, significantly higher than that for the entire domain ($0.09 \pm 1.38\,[ppm]$) and $\sim 45\%$ of the impact of recent surface flux on $X_{\mathrm{CO_2}}$, i.e., $\Delta_{CO_2}^{flux}$.





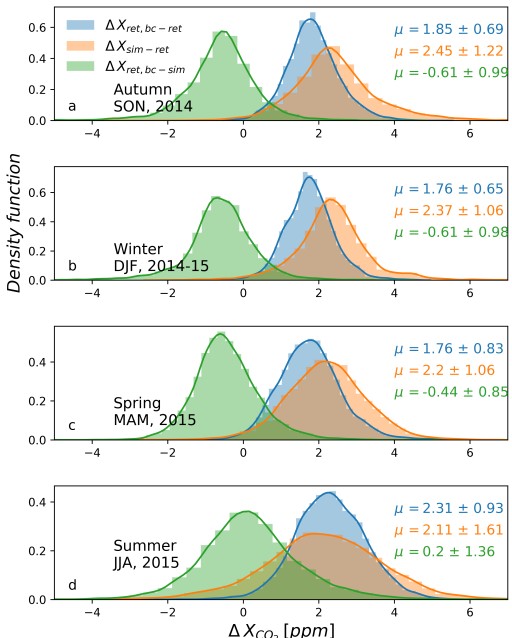

**Figure 3.** Kernel density distributions for residuals between simulated and satellite retrievals, grouped seasonally. Blue distributions show the impact of the OCO-2 bias correction on Land-Nadir retrievals. The orange and green curves show the difference between simulated retrievals and retrievals before and after bias correction respectively. Printed numbers report the mean and standard deviation of residuals. Uncertainties in $X_{CO_2}^{sim}$ are discussed later in sec. 3.3.

### 3.3 Examining bias and uncertainty in $X_{CO_2}^{sim}$

To establish whether differences between $X_{CO_2}^{ret,bc}$ and $X_{CO_2}^{sim}$ presented above (Figs. 3, 4) are due to residual biases in the former, we characterize systemic errors as well errors arising from unresolved variability in model fields used to construct $X_{CO_2}^{sim}$. Cross-model standard deviations of four background models and $\Delta_{CO_2}^{flux}$ from six biospheric flux ensembles on $X_{CO_2}^{sim}$ is shown in Fig. 5. Cross model standard deviation in background fields is highest in the northwest during the summer ($0.36$ $[ppm]$) but usually less than $0.3$ $[ppm]$. Over the entire spatio-temporal domain of the study the standard deviation in estimates of $\Delta_{CO_2}^{flux}$ and $X_{CO_2}^{bkg}$ is $0.26 \pm 0.14$ $[ppm]$ and $0.28 \pm 0.15$ $[ppm]$ respectively. Model spread in $\Delta_{CO_2}^{flux}$ is largest along the pacific coast of Mexico, a region that is relatively less well-constrained by the *in-situ* network. The standard deviation in $\Delta_{CO_2}^{flux}$ is highest in the southeast quadrant in the summer ($0.32$ $[ppm]$), but usually between $0.1$ and $0.3$ $[ppm]$. Uncertainty from model spread in flux and background is propagated in comparisons of $X_{CO_2}^{sim}$ and $X_{CO_2}^{ret,bc}$ presented earlier (Fig. 3).

Systematic error or bias in $X_{CO_2}^{sim}$ can arise from errors in the estimation of background and surface flux, both of which are linked by an atmospheric transport model (Fig. 1). We use the same transport model used by Hu et al. (2019) to generate source-receptor relationships and background (secs. 2.3 and 2.4). Accuracy of the six-member ensemble of fluxes we use is also dependent upon the accuracy of WRF-STILT used in Hu et al. (2019). Thus, potential biases in WRF-STILT form a common



**Figure 4.** Spatial patterns of $X_{CO_2}^{ret,bc}$ (a, e, i, m), $X_{CO_2}^{sim}$ (b, f, j, n), $\Delta X_{sim-ret,bc}$ (c, g, k, o), and impact of recent surface flux ($\Delta_{CO_2}^{flux}$) on the $X_{CO_2}^{sim}$ (d, h, l, p) for soundings grouped seasonally and plotted on a $2° \times 2°$ grid. Units for all maps $[ppm]$. Dotted lines in c, g, k, o r are drawn along 40°N and 100°W.

thread for error propagation. The combined accuracy of fluxes and transport is evaluated by examining aircraft vertical profiles

of $CO_2$ collected under NOAA GML's aircraft program (Sweeney et al., 2015, https://www.esrl.noaa.gov/gmd/ccgg/aircraft/) *not* assimilated by Hu et al. (2019). We simulate all independent aircraft observations over North America for 2007-2015 (the entire spatiotemporal range of Hu et al., 2019) using existing WRF-STILT source-receptor relationships. To ensure consistency with Hu et al. (2019), we perform this evaluation with the same background conditions as in that study (i.e., CT2016). Aircraft profiles are sorted in $1 [km]$ altitude bins from the surface to $8 [km]$ $a.s.l.$ and separated by season. Aircraft vertical profiles

of $CO_2$ (after removing the influence of background) as well as surface fluxes propagated with WRF-STILT show net release





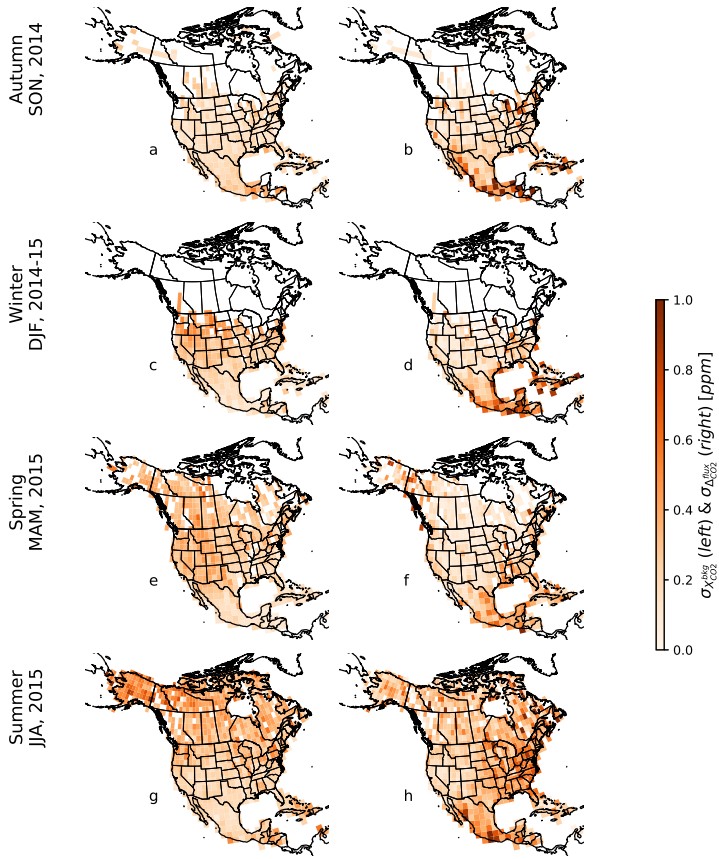

**Figure 5.** Cross model standard deviations flux impact (a, c, e, g) and background models (b, d, f, h). Six biospheric flux ensembles and four model fields for background are used in this study.

of $CO_2$ in non-summer months and net uptake from photosynthesis depleting near-surface $CO_2$ during the summer (green triangles and pink hexagons in Fig. 6a, c, e, g respectively). The difference between independent, unassimilated observations and simulations show that bias at any given altitude level for any season is usually less than $0.5\,[ppm]$ (Fig. 6b, e, h, j). Bias is also usually largest near the surface (except in spring). The pressure-weighted partial column (from the surface to $8\,[km]$ a.s.l.)

mean bias ($\mu_{sim-obs}$) ranges from $-0.12$ in autumn to $0.18\,[ppm]$ in the spring and is comparable to the typical measurement uncertainty within the *in-situ* Global Greenhouse Gasses Research Network of $\sim$ 0.15 [ppm] as derived from long term comparisons of differences between different within-network sampling and analysis approaches for $CO_2$ (e.g., Andrews et al., 2014; Lan et al., 2017; Sweeney et al., 2015). Low partial column bias relative to independent vertical profile $CO_2$ data show that errors in WRF-STLT transport contribute very minimally to bias in $X_{CO_2}^{sim}$.

To examine potential systematic errors in the highest part of the column, (above $8\,[km]$ i.e., the upper troposphere and the stratosphere), we compare 4-D model fields used in this study with $CO_2$ profiles collected by NOAA GML's AirCore program

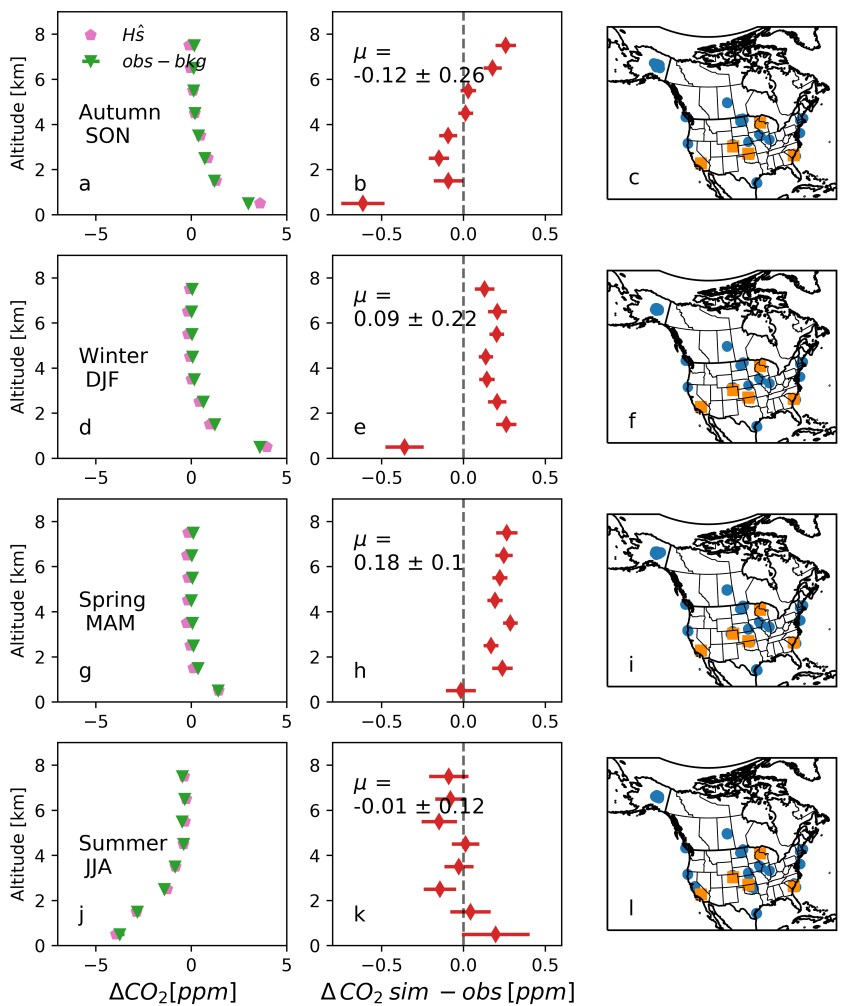

**Figure 6.** Simulated and observed vertical profiles from independent *in-situ* aircraft observations collected over 2007-2015 (a, c, e, g), showing simulated enhancements or depletion in $CO_2$ as a result of surface flux (pink hexagons). Additionally, the difference between observations and CT2016 background is plotted (green triangles). We report the vertically resolved and *pressure weighted* bias (mean and standard deviation) between simulations and observations (b, d, f, h). Finally, a map shows the location of aircraft profiles (blue circles-i) The size of each data point indicates relative number of samples. Additionally, AirCore profile locations are also shown (orange squares-i). AirCore data are used to evaluate biases in boundary conditions in the upper $350\,[hPa]$ of the column and are presented in table 2.





(Karion et al., 2010, data version "v20200210"). There is a considerably larger spread in models' ability to replicate AirCore observations (Table 2) than surface and aircraft observations (Table 1 and Fig. 6). Bias between model and observations varies seasonally, is highest in the spring and lowest in the summer. However, this region forms the upper $\sim 350$ [hPa] of the column.

This has two implications. First, the contribution to total column bias reduces by $\sim 65\%$. Second, the ACOS convolution equation (eq. 1) requires modeled estimates of $X_{CO_2}$ be appropriately weighted with the averaging kernel ($a_i$). $a_i$ determines the balance between information obtained by the retrieval and that contained in the OCO-2 prior ($\chi_{CO_2}^{pri}$), and tends to decrease from $\sim 1$ near the surface to $< 0.5$ in the upper third of the column. Thus $\sim 50\%$ of the error in the highest 350 [$hPa$] of the simulated column is due to errors in $\chi_{CO_2}^{pri}$. Consequently, the total contribution of model bias presented in Table 2 contributes

to only $\sim 50\%$ of the bias in $X_{CO_2}^{sim}$ in the uppermost third of the column. The other source of error above 8 [$km$] is the error in $\chi_{CO_2}^{pri}$, but this information is currently unavailable in the OCO-2 *v10* lite files and is thus ignored in our error analysis. However, errors in $\chi_{CO_2}^{pri}$ are identical for $X_{CO_2}^{sim}$ and $X_{CO_2}^{ret,bc}$ and do not affect comparisons between the two.

**Table 2.** Systematic bias between global 4D $CO_2$ fields and AirCore profiles above 8 [$km$]. Across-model mean and standard deviation is weighted with the pressure weighting function and OCO-2 averaging kernel. All values in [$ppm$]. Location of AirCore profiles is shown as orange squares in Fig. 6i.

| Season | CT 2016 | CT 2019B | CAMS | CarboScope | $\mu_{models} \pm \sigma_{models}$ |
|--------|---------|----------|------|------------|-----------------------------------|
| Autumn | $0.59 \pm 0.02$ | $0.77 \pm 0.01$ | $0.27 \pm 0.01$ | $0.95 \pm 0.02$ | $0.11 \pm 0.04$ |
| Winter | $0.57 \pm 0.01$ | $0.74 \pm 0.01$ | $0.34 \pm 0.01$ | $1.99 \pm 0.02$ | $0.16 \pm 0.19$ |
| Spring | $--$ | $1.40 \pm 0.03$ | $0.76 \pm 0.03$ | $3.31 \pm 0.07$ | $0.32 \pm 0.19$ |
| Summer | $0.4 \pm 0.04$ | $0.45 \pm 0.02$ | $-0.11 \pm 0.02$ | $0.73 \pm 0.03$ | $0.06 \pm 0.05$ |

Combined with the standard deviation across 24 flux-background ensemble members that comprise $X_{CO_2}^{sim}$, comparisons with independent unassimilated aircraft (Fig. 6) and AirCore profiles (Table 2) allow us to comprehensively assess uncertainties

associated with $X_{CO_2}^{sim}$. We estimate the combined uncertainty from surface flux, background estimation and transport as:

$$\sigma_{X_{CO2}^{sim}} = \mu_{model-aircraft} + \mu_{model-AirCore} \pm \sqrt{(\sigma_{model-aircraft})^2 + (\sigma_{model-AirCore})^2 + (\sigma_{flux+bkg\ ensembles})^2} \qquad (5)$$

where

$$\sigma_{flux+bkg\ ensembles} = \sqrt{\sigma_{flux\ ensembles}^2 + \sigma_{bkg\ ensembles}^2 + 2\sigma_{flux,bkg}} \qquad (6)$$

The first two terms in eq. 5 represent the *pressure-weighted* mean partial column bias between modeled and observed $CO_2$

vertical profiles from aircraft (Fig. 6) and aircore (Table 2) respectively. The sum of these two terms provides an estimate of bias or systematic error. The third term in eq. 5 represents unresolved variability.



**Table 3.** Total uncertainty estimates for $X_{CO_2}^{sim}$, the mean difference between $X_{CO_2}^{sim}$ and $X_{CO_2}^{ret,bc}$, and bias in $X_{CO_2}^{ret,bc}$. Bias is obtained as the difference between the first two terms in the middle and left columns. All values in $[ppm]$. The number on the right in the third column indicates the spatial variability in $X_{CO_2}^{ret,bc}$ bias. Finally we also present the same quantity for the previous version of OCO-2 retrievals, the *v9* data product. Note, for *v9* only land-nadir retrievals are analysed.

| Season | $\sigma_{X_{CO2}^{sim}}$ | $\Delta X_{ret,bc-sim}$ | Bias in $X_{CO_2}^{ret,bc}$ | Bias in $X_{CO_2}^{ret,bc}$ *v9-LN* |
|---|---|---|---|---|
| Autumn | $-0.01 \pm 0.31$ | $-0.61 \pm 0.99$ | $-0.62 \pm 0.99$ | $-0.88 \pm 1.00$ |
| Winter | $0.25 \pm 0.35$ | $-0.61 \pm 0.98$ | $-0.36 \pm 0.98$ | $-0.76 \pm 1.10$ |
| Spring | $0.50 \pm 0.31$ | $-0.44 \pm 0.85$ | $0.06 \pm 0.85$ | $0.07 \pm 0.99$ |
| Summer | $0.05 \pm 0.40$ | $0.20 \pm 1.36$ | $0.25 \pm 1.36$ | $0.14 \pm 1.38$ |

We find that unresolved variability due to model spread in $X_{CO_2}^{sim}$ is $\sim 0.35\,[ppm]$ (second term under $\sigma_{X^{sim}}$ in Table 3), similar across seasons, and significantly lower than the uncertainty of $X_{CO_2}^{ret,bc}$ as reported in the OCO-2 *v10 lite* files ($\sim 0.6$ [ppm]). Systematic error or bias in $X_{CO_2}^{sim}$ (first term under $\sigma_{X_{CO2}^{sim}}$ in Table 3) shows significant variability across seasons.

Comparing this to $\Delta X_{ret,bc-sim}$ allows an estimation of mean bias in $X_{CO_2}^{ret,bc}$ over North America (Table 3). We find that $X_{CO_2}^{ret,bc}$ bias ranges from $-0.62\,[ppm]$ in autumn to $0.14\,[ppm]$ in summer. This indicates that $\Delta X_{ret,bc-sim}$ in autumn (Fig. 3a and Fig. 4c) is entirely due to a residual bias in $X_{CO_2}^{ret,bc}$ but almost entirely due to bias in $\sigma_{X_{CO2}^{sim}}$ in the spring (Fig. 3c and Fig. 4k). During summer the mean bias in $X_{CO_2}^{sim}$ is $0.05\,[ppm]$, and consequently the mean bias in $X_{CO_2}^{ret,bc}$ is $0.25\,[ppm]$. However, the mean conceals larger regional differences. In the northeast quadrant for instance, $\Delta X_{ret,bc-sim}$ (Fig. 4o) of

$0.79\,[ppm]$ translates to a high bias $0.84\,[ppm]$ during the summer. Finally, we find that the OCO-2 *v10* bias correction shows an improvement over the *v9* bias correction, particularly in the winter.

### 3.4   Evaluating the OCO-2 bias correction

Bias in OCO-2 *v10* $X_{CO_2}$ is a combination of footprint bias (8 coincident across track retrievals; $C_f$) and feature biases (related to surface or atmospheric parameters, e.g., aerosol optical depth; $C_p$). Finally, a global scaling factor ($C_0$) obtained

from comparisons with TCCON retrievals, is used to empirically link retrievals to the WMO scale.

To examine residual feature biases, we perform simple linear regressions between parameters used in the OCO-2 *v10* bias correction with $\Delta X_{ret,bc-sim}$. These parameters are $\Delta P_{frac}\,[ppm]$, which accounts for fractional change in $X_{CO_2}$ due to difference in prior and retrieved surface pressure (Kiel et al., 2019), $CO_2\text{-}grad\,del$, defined as the difference of the difference between retrieved and prior $CO_2$ at the surface and at 0.7 times the surface pressure, $dws$, which is the total retrieved optical

depth associated with aerosols from dust, water cloud and aerosol and $aod_{fine}$, the aerosol optical depth from sulfate and organic carbon (O'Dell et al., 2018; Osterman et al., 2020). Additionally, we perform simple linear regressions with altitude and surface albedo. All parameters are available in the OCO-2 *v10 lite* files. We find no significant correlations observed





between $\Delta\Delta X_{ret,bc-sim}$ and any parameters suggesting that there are no regional scale parametric biases over North America for our study period that are not already removed by the OCO-2 global bias correction.

We then evaluate the OCO-2 *v10* global scaling by obtaining scaling factors ($C_{0,sim}$) for North America. First, we 'de-link' $X_{CO_2}^{ret,bc}$ from TCCON by multiplying it with $C_0$. Then we invert the framework defined by O'Dell et al. (2018) and obtain $C_{0,sim}$ as the slope of a linear regression between $X_{CO_2}^{sim}$ and $X_{CO_2}^{ret,bc}$ that is forced through the origin, with associated uncertainties (York et al., 2004). $X_{CO_2}^{sim}$ uncertainty is characterized as the standard deviation of all background and flux ensemble members described in 2.4 and 2.5. We also propagate both systematic errors and residual errors due to unresolved

variability in $X_{CO_2}^{sim}$ (sec. 3). We use the reported uncertainty of each retrieval as uncertainty in $X_{CO_2}^{ret}$. $C_{0,sim}$ varies seasonally for 2014-15 and is also different from the static, global, multi-year number used for land-nadir retrievals currently (Table 4). To test the impact of this difference, we scale $X_{CO_2}^{ret,bc}$ with $C_{0,sim} - C_0$ for each season. The mean $\Delta X_{CO_2}$ [ppm] for all retrievals is presented in the third column in table 4, and indicates the magnitude by which $X_{CO_2}^{ret,bc}$ would have to change so as to be consistent with $X_{CO_2}^{sim}$., i.e. the *in-situ* network over North America. For e.g., if instead of the TCCON network, $X_{CO_2}^{sim}$ is

used to scale land-nadir OCO-2 *v10* soundings over North America, $X_{CO_2}^{ret,bc}$ would reduce by $-0.61$ [ppm] in autumn.

**Table 4.** Scaling factors for global bias obtained from $X_{CO_2}^{sim}$ ($C_{0,sim}$) and the global number used in the OCO-2 bias correction ($C_0$). $C_{0,sim}$ varies seasonally while $C_0$ is constant. The error in $C_{0,sim}$ is $< 1.0e^{-4}$ for all regressions. $\Delta X_{CO_2}$ is calculated by multiplying the difference between $C_{0,sim}$ and $C_0$ with the mean $X_{CO_2}^{ret,bc}$ for each season and indicates the magnitude by which $X_{CO_2}^{ret,bc}$ would have to adjust if $X_{CO_2}^{sim}$ is used in global scaling, instead of TCCON.

| Season | $C_{0,sim}$ | $C_0$ | $\Delta X_{CO_2}$ [ppm] |
|---|---|---|---|
| Autumn | 0.9943 | | $-0.61$ |
| Winter | 0.9950 | | $-0.36$ |
| Spring | 0.9960 | | 0.06 |
| Summer | 0.9965 | | 0.24 |
| Annual | 0.9960 | 0.9959 | 0.24 |

## 4   Implications for carbon flux estimation

The impact of recent surface flux on $X_{CO_2}^{sim}$ is small (e.g. right column in Fig. 4). The interquartile range of $\Delta_{CO_2}^{flux}$ over the entire spatiotemporal domain is less than 1 [ppm] implying that the imprint of recent surface flux on the total column is roughly half the magnitude of the OCO-2 bias correction (blue curves in Fig.3). Moreover, only around 2 % of simulations in

the summer (when surface fluxes are highest) are associated with absolute $\Delta_{CO_2}^{flux}$ magnitudes higher than 4 [ppm], indicating that recent surface flux rarely accounts for more than a $\sim 1$ % change in $X_{CO_2}^{sim}$. In autumn, $\Delta_{CO_2}^{flux}$ is of the same magnitude as bias in $X_{CO_2}^{ret,bc}$ (Table 3). OCO-2 retrievals in autumn 2014 are therefore unlikely to provide reliable estimates of North

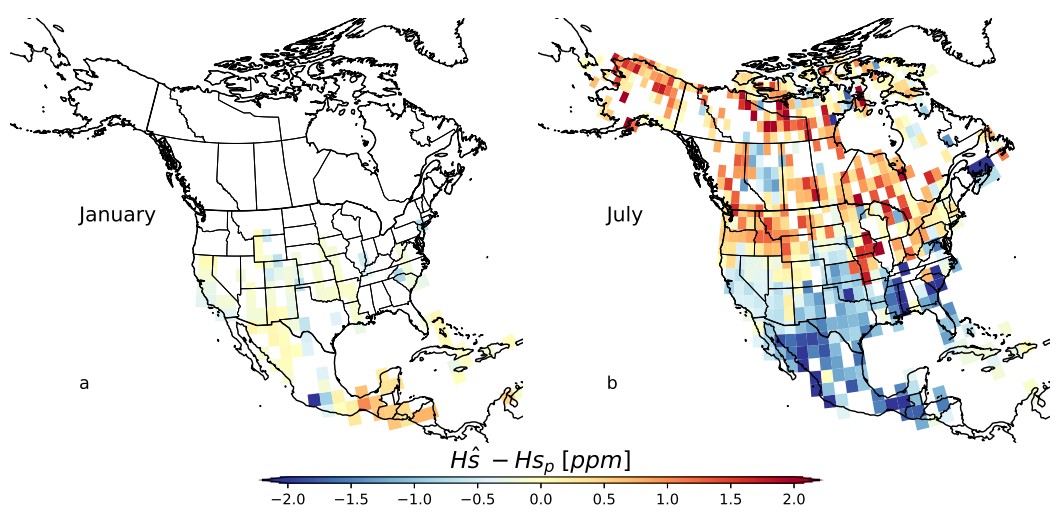

**Figure 7.** Difference between prior and optimized biospheric flux from CarbonTracker Lagrange show small differences, when projected onto the total column [ppm].

American surface flux. During the summer, OCO-2 has a high bias of $0.25\,[ppm]$ over the continent, but this bias may be significantly larger in the eastern half of the domain (Fig. 4o). In the northeast quadrant for example, a potential bias of $0.84\,[ppm]$ constitutes $\sim 47$ % of the mean $\Delta_{CO_2}^{flux}$ of $-1.79\,[ppm]$. Considering that the vast majority of current inverse modeling or data assimilation systems used for $CO_2$ flux estimation are designed to correct errors in an *a priori* estimate, the effective signal is considerably smaller than what is shown here (right column in Fig. 4). In fact, when projected onto the total column, we find that the difference between a biospheric prior flux model (CASA-CMS) and flux optimized by the CT-Lagrange inversion system (using the same prior flux estimate; Hu et al., 2019) for January and July, 2015 are $0.15 \pm 0.38$ and $0.09 \pm 1.06\,[ppm]$ respectively. Continental scale aggregations mask regional differences (Fig. 7), but flux adjustment impacts on $X_{CO_2}^{sim}$ are usually indistinguishable from 0 in January and greater than $\pm 1\,[ppm]$ for less than a third of simulated retrievals in July, when biospheric uptake over North America is highest. Errors in terrestrial biosphere models of $CO_2$ flux translate to similarly small impacts on $X_{CO_2}$, posing questions on the utility of these data currently in evaluating terrestrial biosphere models.

# 5 Conclusions

In this study, we compare one year of $X_{CO_2}^{ret,bc}$ over North America from NASA's OCO-2 (*v10*; *land-nadir* and *glint* retrievals) satellite against synthetic columns that are constructed using a high-resolution regional model of atmospheric transport and


driven by fluxes and background that are optimally consistent with *in-situ* measurements of $CO_2$ dry air mole fraction, which are rigorously calibrated to the WMO $CO_2$ X2007 scale. Although $X_{CO_2}$ from OCO-2 and from the posterior of *in-situ* data

inversions has been compared previously, and used in its bias correction (O'Dell et al., 2018; Kiel et al., 2019), this is the first such evaluation at the regional scale that uses high-resolution atmospheric transport. We use a suite of optimized non-fire net ecosystem exchange fluxes and background fields to assess under in $X_{CO_2}^{sim}$. Potential systematic errors in fluxes, transport, and background fields are evaluated by comparisons with vertical gradients of atmospheric $CO_2$ from independent aircraft and AirCore vertical profiles. $X_{CO_2}^{sim}$ is associated with errors arising from unresolved variability in model fields, and systematic

bias. The first of these results in an uncertainty of $\sim 0.35$ [ppm]. Bias or systematic error in $X_{CO_2}^{sim}$ is found to vary seasonally and ranges from $-0.01$ [ppm] in autumn to $0.50$ [ppm] in the spring. Bias is highest in the upper \$ 350 [h Pa] of the column (Table 2), a region that is most poorly constrained by atmospheric measurements. However, the effect of this bias is relatively small in the total column comparisons.

Comparisons with $X_{CO_2}^{sim}$ show that the OCO-2 *v10* global bias correction greatly improves the quality of OCO-2 data over

North America (Fig. 3). However, generally good agreement between $X_{CO_2}^{ret,bc}$ and $X_{CO_2}^{sim}$ at the continental scale masks significant differences at regional scales and for some seasons (Fig. 4). Error analysis of the components of $X_{CO_2}^{sim}$ (i.e., transport, background, and flux) allows us to better characterize difference between simulations and retrievals. Differences in $\Delta X_{\mathrm{ret,bc-sim}}$ are highest in autumn and indicative of a low bias in $X_{CO_2}^{ret,bc}$ of $0.62$ [ppm] which is identical to the mean impact of recent surface flux (mean $\Delta_{CO_2}^{flux}$ over the continent is $0.64$ [ppm]) in that season. In winter, a low bias in $X_{CO_2}^{ret,bc}$

of $0.36$ [ppm] is roughly $50\%$ of the mean $\Delta_{CO_2}^{flux}$ of $0.71$ [ppm]. In summer we find spatially coherent regional patterns in $\Delta X_{\mathrm{ret,bc-sim}}$. $\Delta X_{\mathrm{ret,bc-sim}}$ is highest in the northeast quadrant of North America (Fig. 4o) at $-0.81$ [ppm], $50\%$ of the mean expected ecosystem flux impact over this region. Since inverse models of $CO_2$ flux usually optimize a prior flux estimate, the surface flux signal (i.e., difference between prior and optimized flux) in $X_{CO_2}$ is minuscule (Fig. 7), significantly smaller than the magnitude of the OCO-2 *v10* bias correction, and translates to extremely strenuous requirements on the quality of

space-based retrievals. The OCO-2 community has worked diligently to reduce uncertainty on satellite retrievals (e.g., O'Dell et al., 2012, 2018; Kiel et al., 2019; Wunch et al., 2017; Kulawik et al., 2019); bias in *v10* retrievals over North America has reduced in autumn and winter of 2014-15 compared to the *v9* data product (Table 3), but further improvement is necessary for both existing satellite datasets and planned missions that will provide this quantity in order to accurately constrain surface fluxes in a changing climate. Finally, we argue that a greatly expanded global reference network of calibrated *in-situ* vertical

profile measurements is necessary to reliably detect and correct systematic errors in satellite $X_{CO_2}$.

*Competing interests.* No competing interests are present.

*Acknowledgements.* This work was funded by NASA CMS grant: Guan (CMS 2016): Improving the monitoring capability of carbon budget for the US Corn Belt- integrating multi-source satellite data with improved land surface modeling and atmospheric inversion to KG, and



NASA CMS grant: Andrews (CMS 2016): Regional Inverse Modeling in North and South America for the NASA Carbon Monitoring System to AEA. OCO-2 *v10* data were obtained from NASA Goddard Earth Science Data and Information Services Center (GES DISC) https://disc.gsfc.nasa.gov/datasets?page=1&keywords=OCO-2, last accessed on April 7, 2021). These data were produced by the OCO-2 project at the Jet Propulsion Laboratory, California Institute of Technology, and obtained from the OCO-2 data archive maintained at the NASA Goddard Earth Science Data and Information Services Center. NOAA AirCore profiles were provided by Bianca Baier and Colm Sweeney (GML, NOAA; https://www.esrl.noaa.gov/gmd/ccgg/aircore/, last accessed on April 7, 2021). Thanks to Sydnee Masias for help with the conceptual figure and Chris O'Dell for comments on a previous version of the manuscript.



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
