# Peer review of "Evaluating consistency between total column $CO_2$ retrievals from OCO-2 and the *in-situ* network over North America: Implications for Carbon flux estimation"

_Atmospheric Chemistry and Physics, 2021_

## Referee Comment (RC1)

**Referee Report:**
**Evaluating consistency between total column $CO_2$ retrievals from OCO-2 and the *in-situ* network over North America: Implications for carbon flux estimation**

Anonymous

**1 Overview**

In this study, Rastogi et al. reports comparisons of total column $CO_2$ retrieved from NASA's OCO-2 satellite ($X_{CO_2}^{ret}$) and constructed using a high-resolution regional model ($X_{CO_2}^{sim}$) over North America. The manuscript is very clear and well presented. This manuscript is well within the scope of ACP. I recommend that the manuscript be published in ACP after minor revision.

**2 Minor comments**

(1) Page 5, Line 119: it is described in the text that $\Delta_{CO_2,i}^{flux}$ is computed at discrete levels from the surface to 14 km, does this indicate that level $N-3$ in equation (3), which is the top level used for $\Delta_{CO_2,i}^{flux}$, corresponds to 14 km? Please clarify.

(2) Page 16: 3.4 Evaluating the OCO-2 bias correction: in examination of residual feature biases, the authors state that there is no significant correlations between $\Delta X_{ret,bc-sim}$ and the listed parameters. However, no data is provided. It would be more informative/quantitative to show at least some typical results of the linear regressions.

**3 Technical corrections**

(1) Page 3, Line 81: "[ppm]" $\rightarrow$ "[ppm (parts per million dry air mole fraction)]" (Andrews et al., 2014).

(2) Page 5, Line 107: if possible, please provide the version of ACOS used in this study.

(3) Page 9, Line 223: "NOAA's CarbonTracker-Lagrange": please provide the version of this model which is used to carry out the simulations in this study.

(4) Page 11, Line 262: "systemic errors as well errors" $\rightarrow$ "systemic errors as well as errors".

(5) Page 12, Line 279: "a.s.l." $\rightarrow$ "a.s.l. (above sea level)".

(6) Page 17, Line 333: "$\Delta\Delta X_{ret,bc-sim}$" $\rightarrow$ "$\Delta X_{ret,bc-sim}$".

(7) Page 17, Table 4: the value of $C_0$ is missing for each season. If the value of $C_0$ is constant as stated in the text, please fill in this value for each season. Otherwise it may cause confusion to the readers.

(8) Page 19, Line 376: "upper $ 350 [h Pa]" $\rightarrow$ "upper 350 [hPa]"

**References**

Andrews, A. E., Kofler, J. D., Trudeau, M. E., Williams, J. C., Neff, D. H., Masarie, K. A., Chao, D. Y., Kitzis, D. R., Novelli, P. C., Zhao, C. L., Dlugokencky, E. J., Lang, P. M., Crotwell, M. J., Fischer, M. L., Parker, M. J., Lee, J. T., Baumann, D. D., Desai, A. R., Stanier, C. O., De Wekker, S. F. J., Wolfe, D. E., Munger, J. W., and Tans, P. P.: $CO_2$, CO, and $CH_4$ measurements from tall towers in the NOAA Earth System Research Laboratory's Global Greenhouse Gas Reference Network: instrumentation, uncertainty analysis, and recommendations for future high-accuracy greenhouse gas monitoring efforts, Atmospheric Measurement Techniques, 7, 647–687, https://doi.org/10.5194/amt-7-647-2014, https://amt.copernicus.org/articles/7/647/2014/, 2014.

---

## Referee Comment (RC2)

The authors describe a method wherein in situ-constrained $XCO_2$ simulations over North America are used to evaluate retrievals of $XCO_2$ from OCO-2 v10. The paper could be appropriate for publication after several major concerns are addressed.

Major comments:

On lines 120 and 194 and elsewhere, the authors state their assumption that the influence of recent surface fluxes are zero in the upper troposphere and stratosphere. This is not the case during some forest fires (e.g., Hooghiem et al., 2020 and references therein) and some volcanic eruptions. How does this assumption influence the validity of the results, and in particular, the seasonal cycle? How realistic is the background $XCO_2$ at those altitudes? The satellites measure through the entire atmosphere, and even if the averaging kernels are smaller in the stratosphere than in the troposphere, they are nonzero – that region of the atmosphere will still influence the retrievals. The authors discuss the errors associated with this around line 295 and dismiss the differences as small, but they do not show profiles comparing the simulated profiles and AirCore above 8 km.

On lines 335-345, the authors evaluate the OCO-2 scaling to the WMO X2007 scale using a single year (2014-2015) and their simulated $XCO_2$ product. Seasonal scaling does not make sense in this context, as the scaling is largely required because of spectroscopic uncertainties, and it is therefore more likely to be related to the airmass or water vapour interference than to the season. I suggest that these results are checked by binning the C0 values over different airmasses instead of season. The authors also claim that their annual C0,sim (0.9960) is inconsistent with the C0 derived through comparisons with TCCON (0.9959), but given the different timescales and spatial scales used in the analysis, I do not think this is a robust conclusion.

The third and fourth paragraphs of the introduction betray the authors' low opinion of the value of remote sensing relative to in situ measurements. I strongly recommend that the authors completely rewrite those paragraphs as they are misleading. I shall respond to each sentence of these two paragraphs in turn (original text in *red italics*):

*In-situ measurements that comprise global networks, such as NOAA's Global Greenhouse Reference Network (https://www.esrl.noaa.gov/gmd/ccgg/ggrn.php), are rigorously evaluated and carefully calibrated relative to the World Meteorological Organization (WMO) calibration scale (data used here are reported on the X2007 scale), thus ensuring the fidelity of these measurements over timescales of seasons to decades (Andrews et al., 2014; Hall et al., 2020).*

I agree that in situ measurements like NOAA's Reference Network are rigorously evaluated and calibrated. While the authors do cite the (now published) Hall et al. 2021 paper, it seems relevant to explicitly mention that these calibrations are complex and that periodic and significant changes of scale are possible. From the Hall et al. 2021 paper (italics are my additions): "The new scale *[WMO-CO2-X2019]* is 0.18 $\mu$mol mol$^{-1}$ (ppm) greater than the previous scale *[WMO-CO2-X2007]* at 400 ppm $CO_2$. While this difference is small in relative terms (0.045 %), it is significant in terms of atmospheric monitoring."

*The "open-path" nature of space-based XCO₂ measurements however, does not allow for direct calibration.*

I agree with this sentence but feel that the authors' use of "however" is unnecessary.

*Satellite retrievals of XCO2 require complicated models of atmospheric radiation and are sensitive to a host of assumptions about aerosols, clouds, interference of jointly retrieved parameters, surface properties and details of the instrumentation (Kulawik et al., 2019).*

Again, I largely agree with the sentiment, but the language betrays a value judgment ("complicated," "host of assumptions"). There has been significant research into these areas.

*Moreover, sensors typically degrade over time, and limited information is available to characterize resulting time-dependent systematic errors.*

Sensors can and do degrade over time, but there is sufficient data available to characterize time-dependent systematic errors through radiometric calibrations, lunar calibrations, and comparisons with ground-based data (e.g., Crisp et al., 2017; Bruegge et al., 2019; Yu et al., 2020).

*Post-launch data corrections are performed if and when biases (O'Dell et al., 2018) and errors (Kiel et al., 2019) are identified, but are severely limited due to the sparsity of calibrated in-situ vertical profile observations.*

Post-launch data corrections are rigorously and systematically evaluated against the available ground-based information provided by multiple sources (TCCON, models, small area analyses, southern hemisphere approximation, etc.). None of this is limited by "the sparsity of calibrated in-situ vertical profile observations." This is not done "if and when biases and errors are identified" – they are actively sought out and significant effort is put into this.

*Currently, satellite derived XCO2 retrievals are linked to the WMO scale most directly through a limited set of in-situ profiles obtained over a network of ground-based Fourier Transform Infrared Spectrometers that comprise the Total Carbon Column Observation Network (TCCON; Wunch et al., 2017).*

Satellite retrievals of XCO₂ are linked to the WMO scale through thousands of coincident measurements over TCCON stations throughout the lifetime of the mission.

*However, TCCON itself provides remotely sensed information about XCO2, and TCCON retrievals undergo a complex validation and bias correction routine (Wunch et al., 2010) that links these retrievals to the WMO scale.*

The TCCON data have been rigorously and systematically compared with in situ profiles that are, in turn, calibrated to the WMO X2007 scale. There have been about 80 such CO₂ in situ profiles over TCCON stations – the first collected in 2004, and the most recent from the current year. I do not know what the authors mean by "TCCON retrievals undergo a complex

validation." The paper cited details the method of tying the TCCON retrievals to the WMO X2007 scale.

*Moreover, TCCON sites are few.*

There are 26 currently operating TCCON stations.

*Issues with validation of OCO-2 retrievals via TCCON have been identified- seasonal and site-dependent biases have been reported (Wunch et al., 2017, 2015), raising questions about the adequacy of this network to validate satellite derived XCO2 products (Basu et al., 2013).*

I do not see how the 2015 paper (describing the GGG2014 algorithm and dataset) is related to this topic. The 2017 paper identifies biases in OCO-2, and not in the TCCON data – the 2013 paper discusses biases in the RemoTeC GOSAT retrievals relative to TCCON. I do not think these papers question the "adequacy" of the network to provide valuable validation information.

*This is especially important since systematic bias corrections of XCO2 from TCCON data are developed over small spatio-temporal scales and extrapolated globally (Wunch et al., 2011) for retrievals over land and ocean respectively (O'Dell et al., 2018).*

I do not understand what the authors are referring to in this statement. If the authors are referring to the systematic bias identification related to retrieval parameters like albedo, aerosol, etc., then I do not know what "small spatio-temporal scales" the authors are referring to. The 2011 work did not use TCCON data to identify these systematic biases – it used GOSAT data from the southern hemisphere south of 25S. Comparisons with the TCCON data showed that the bias correction improved the comparisons globally, and identified the scaling required to tie the bias-corrected GOSAT $XCO_2$ to the WMO scale.

*OCO-2 retrievals are additionally corrected for bias by comparing with 4-D CO2 mole fraction fields from global inverse models, and a small-area approximation, but both methods are prone to smoothing across fine-scale variability in XCO2 (O'Dell et al., 2018; Corbin et al., 2008).*

I don't understand this statement at all. It should be *comforting* that bias corrections derived from these independent methods are consistent.

*While bias correction generally reduces inferred surface flux uncertainty when retrievals are assimilated in atmospheric inversions (Basu et al., 2013), even small retrieval errors can lead to large errors in inferred flux (Takagi et al., 2014; Chevallier et al., 2014).*

I agree.

*Biases in XCO2 from OCO-2, hereafter XretCO2 , have been identified, and found to be related to surface (e.g. pressure, albedo) and atmospheric (e.g. aerosol loading, sky condition) properties (Kiel et al., 2019).*

The Kiel et al., 2019 paper talks about a geolocation error and a meteorological reanalysis sampling error. I'm not sure why the authors attribute aerosol, albedo, cloud cover to this paper.

*However, systematic biases not accounted for in the v10 bias correction approach persist and therefore the measurement uncertainty associated with individual retrievals is believed to be at least twice the value currently reported in the OCO-2 data files (Eldering et al., 2017).*

I don't see how the authors can justify this claim with the citation provided. Version 10 was released in 2020, and the Eldering paper is dated in 2017.

*Thus, a dynamic method to routinely evaluate satellite retrievals is necessary.*

This statement does not follow from the previous two paragraphs. Define what is meant by "a dynamic method".

Other comments:

Figure 7 and lines 360-364: There's a 4-ppm latitudinal gradient in the column across North America before and after optimizing biospheric fluxes. This is not "small" from a carbon cycle perspective and should be detectable from space.

**References not already cited in the paper:**

Hooghiem, J. J. D., et al.: Wildfire smoke in the lower stratosphere identified by in situ CO observations, Atmos. Chem. Phys., 20, 13985–14003, https://doi.org/10.5194/acp-20-13985-2020, 2020.

Hall, B. D., et al.: Revision of the World Meteorological Organization Global Atmosphere Watch (WMO/GAW) $CO_2$ calibration scale, Atmos. Meas. Tech., 14, 3015–3032, https://doi.org/10.5194/amt-14-3015-2021, 2021.

Crisp, D., et al.: The on-orbit performance of the Orbiting Carbon Observatory-2 (OCO-2) instrument and its radiometrically calibrated products, Atmos. Meas. Tech., 10, 59–81, https://doi.org/10.5194/amt-10-59-2017, 2017.

C. J. Bruegge et al.: Vicarious Calibration of Orbiting Carbon Observatory-2, in *IEEE Transactions on Geoscience and Remote Sensing*, vol. 57, no. 7, pp. 5135-5145, July 2019, doi: 10.1109/TGRS.2019.2897068.

Yu S, et al.: Stability Assessment of OCO-2 Radiometric Calibration Using Aqua MODIS as a Reference. *Remote Sensing*. 2020; 12(8):1269. https://doi.org/10.3390/rs12081269

---

## Author Comment (AC1)

**Response to Reviewer:**
We thank the reviewer for a careful review of our manuscript. Please see our responses below:

**Responses to minor comments:**

2.1. We have clarified the text as follows:

" $\Delta_{CO_2}^{flux}$ is computed at discrete levels from the surface to 14 [km], whereas $\chi_{CO_2}^{bkg}$ is computed at 3 additional levels. These additional levels represent the upper troposphere and the stratosphere, where influence of recent surface flux is assumed to be zero. If there are cases where recent surface fluxes influence upper tropospheric and stratospheric air, those are accounted for as part of background estimation. This is because models used to estimate background are also constrained by *in-situ* measurements. Note that there may still be rare cases, e.g., in the proximity of large fires (Hooghiem et al., 2020), where surface flux influence in the upper troposphere and lower stratosphere may not be captured by this approach."

2.2. We have now included these regressions as supplementary material S1.

**Responses to technical corrections:**

3.1. ppm acronym has been spelled out as per the reviewer's suggestion.

3.2. We have added the ACOS retrieval version (v10).

3.3. NOAA's CarbonTracker-Lagrange model is not versioned currently, as there is only one version (https://gml.noaa.gov/ccgg/carbontracker-lagrange/). This is a regional Lagrangian model that is different from the global Eulerian version (NOAA's CarbonTracker; https://gml.noaa.gov/ccgg/carbontracker/) that is released quasi-annually.

3.4. We have corrected 'systemic' to 'systematic'.

3.5. a.s.l. has now been changed to a.s.l. (above sea level).

3.6. We have amended this error as per the reviewer's suggestion.

3.7. Considering the other reviewer's comments, we have now removed this section of the manuscript.

3.8. This has been corrected.

---

## Author Comment (AC2)

The authors describe a method wherein in situ-constrained $XCO_2$ simulations over North America are used to evaluate retrievals of $XCO_2$ from OCO-2 v10. The paper could be appropriate for publication after several major concerns are addressed.

We thank the reviewer for their careful review of the manuscript. We have significantly revised two paragraphs (paragraphs 3 and 4, lines 67-76 in the introduction that were flagged for concern by the reviewer. For each paragraph, we present the revised text (in quotes) below all comments for that paragraph in the response below.

Additionally, we have revised the preceding paragraph (paragraph 2, lines 51-66) to clarify the intent of our analysis: To better understand the use of satellite data for regional flux estimation.
Finally, we believe we have addressed all additional comments and concerns, as outlined in detail below.
As a general note, we wish to clarify that the authors are not putting forth an opinion of satellite retrievals relative to in-situ data. We have taken great care in this work to put forth quantitative and objective analysis. As such, we have, in several places throughout the paper, edited the text for clarity in this regard. We hope that these changes address the reviewer's overall impression that this manuscript offers "opinions" in any form. This was not the intent of the paper as written and we believe the below revisions aid the clarity of the message.
To summarize this message: the satellite community continues to make great advances in theoretical understanding (spectroscopy), instrument design, and evaluation of bias. Even compared with simulations in this study, potential bias in v10 is lower than that for v9 (Table 4 in the revised text). The purposes of this manuscript are, however, to assess the specific use case of satellite data for inferring regional scale $CO_2$ fluxes. We probe three critical points. First, terrestrial net non-fire biospheric flux (NEE) is a balance of two opposing fluxes (Gross Primary Productivity and Ecosystem Respiration) and even when these one-way fluxes are large, the balance can be very small. Second, NEE occurs everywhere on land unlike spatially discrete point sources from industrial emissions. Finally, $CO_2$ has a long lifetime in the atmosphere and therefore is well-mixed. Together, these points mean that $CO_2$ mole fraction enhancements or depletions due to NEE over a given region can be very small and hard to distinguish from variability in $CO_2$ mole fraction resulting from upwind fluxes and transport. The problem is magnified in the total column, as NEE signals at the surface are diluted over the path length of the atmosphere. For the purposes of assessing NEE fluxes with satellite data, it is therefore a critical exercise to understand and assess the accuracy of that data, which is what this manuscript aims to achieve.

Major comments:

On lines 120 and 194 and elsewhere, the authors state their assumption that the influence of recent surface fluxes are zero in the upper troposphere and stratosphere. This is not the case during some forest fires (e.g., Hooghiem et al., 2020 and references therein) and some volcanic eruptions. How does this assumption influence the validity of the results, and in particular, the seasonal cycle? How realistic is the background $XCO_2$ at those altitudes? The satellites measure through the entire atmosphere, and even if the averaging kernels are smaller in the stratosphere

than in the troposphere, they are nonzero – that region of the atmosphere will still influence the retrievals. The authors discuss the errors associated with this around line 295 and dismiss the differences as small, but they do not show profiles comparing the simulated profiles and AirCore above 8 km.

We sample global Eulerian model fields (optimized with in-situ observations calibrated on the WMO scale) to estimate background and present seasonal comparisons of all background models with independent aircore observations above 8 km (Table 2) and propagate those errors when comparing simulated $X_{CO2}$ with OCO-2 $X_{CO2}$. Since background models are optimized by in-situ observations, these should be able to capture those surface fluxes to the extent that they impact the observations. While relatively coarse global models may not be able to capture individual events such as fire very well, in analyzing thousands of aircraft profiles of CO and $CO_2$ over North America collected over the last 10-15 years, the incidence of fire signals in the mid and upper troposphere is very low. To clarify this point for the reader, we have added he following at line 119 of the revised text:

"If there are cases where recent surface fluxes influence upper tropospheric and stratospheric air, those are accounted for as part of background estimation. This is because models used to estimate background are also constrained by in-situ measurements. Note that there may still be rare cases, e.g., in the proximity of large fires (Hooghiem et al., 2020), where surface flux influence in the upper troposphere and lower stratosphere may not be captured by this approach."

On lines 335-345, the authors evaluate the OCO-2 scaling to the WMO X2007 scale using a single year (2014-2015) and their simulated $XCO_2$ product. Seasonal scaling does not make sense in this context, as the scaling is largely required because of spectroscopic uncertainties, and it is therefore more likely to be related to the airmass or water vapour interference than to the season. I suggest that these results are checked by binning the C0 values over different airmasses instead of season. The authors also claim that their annual C0,sim (0.9960) is inconsistent with the C0 derived through comparisons with TCCON (0.9959), but given the different timescales and spatial scales used in the analysis, I do not think this is a robust conclusion.

The reviewer has highlighted that this conclusion may not be robust given the available data. In response, we have removed this section from the manuscript.

The third and fourth paragraphs of the introduction betray the authors' low opinion of the value of remote sensing relative to in situ measurements. I strongly recommend that the authors completely rewrite those paragraphs as they are misleading. I shall respond to each sentence of these two paragraphs in turn (original text in *red italics*):

Note that, in response to the reviewer's comments, for each of the two paragraphs in question, we present the revised text (in quotes) below all comments for that paragraph below.

*In-situ measurements that comprise global networks, such as NOAA's Global Greenhouse Reference Network (https://www.esrl.noaa.gov/gmd/ccgg/ggrn.php), are rigorously evaluated and carefully calibrated relative to the World Meteorological Organization (WMO) calibration scale (data used here are reported on the X2007 scale), thus ensuring the fidelity of these*

*measurements over timescales of seasons to decades (Andrews et al., 2014; Hall et al., 2020).*

I agree that in situ measurements like NOAA's Reference Network are rigorously evaluated and calibrated. While the authors do cite the (now published) Hall et al. 2021 paper, it seems relevant to explicitly mention that these calibrations are complex and that periodic and significant changes of scale are possible. From the Hall et al. 2021 paper (italics are my additions): "The new scale *[WMO-CO2-X2019]* is 0.18 μmol mol$^{-1}$ (ppm) greater than the previous scale *[WMO CO2-X2007]* at 400 ppm $CO_2$. While this difference is small in relative terms (0.045 %), it is significant in terms of atmospheric monitoring."

*The "open-path" nature of space-based $XCO_2$ measurements however, does not allow for direct calibration.*

I agree with this sentence but feel that the authors' use of "however" is unnecessary.

*Satellite retrievals of XCO2 require complicated models of atmospheric radiation and are sensitive to a host of assumptions about aerosols, clouds, interference of jointly retrieved parameters, surface properties and details of the instrumentation (Kulawik et al., 2019).*

Again, I largely agree with the sentiment, but the language betrays a value judgment ("complicated," "host of assumptions"). There has been significant research into these areas.

*Moreover, sensors typically degrade over time, and limited information is available to characterize resulting time-dependent systematic errors.*

Sensors can and do degrade over time, but there is sufficient data available to characterize time dependent systematic errors through radiometric calibrations, lunar calibrations, and comparisons with ground-based data (e.g., Crisp et al., 2017; Bruegge et al., 2019; Yu et al., 2020).

*Post-launch data corrections are performed if and when biases (O'Dell et al., 2018) and errors (Kiel et al., 2019) are identified, but are severely limited due to the sparsity of calibrated in-situ vertical profile observations.*

Post-launch data corrections are rigorously and systematically evaluated against the available ground-based information provided by multiple sources (TCCON, models, small area analyses, southern hemisphere approximation, etc.). None of this is limited by "the sparsity of calibrated in-situ vertical profile observations." This is not done "if and when biases and errors are identified" – they are actively sought out and significant effort is put into this.

Revised text # 1

"Satellite retrievals of total column $CO_2$ mole fraction ($X_{CO2}$), such as those from NASA's Orbiting Carbon Observatory-2 (OCO-2), have the potential to provide unprecedented information about spatio-temporal patterns and variability in the Earth's atmosphere.

However, observations of $X_{CO2}$ must be extremely stable, highly precise and free from bias to detect and quantify abundance changes caused by a change in surface fluxes (Rayner and O'Brien 2001; Olsen et al., 2004, Miller et al., 2007, Houweling et al., 2003). Regional flux of terrestrial net non-fire ecosystem exchange of $CO_2$ (NEE) can be small as it is composed of two opposing fluxes (photosynthesis and respiration). Further, NEE is ubiquitous on the terrestrial surface (unlike for e.g., spatially discrete point sources from industrial emissions). Lastly, $CO_2$ has a long lifetime in the atmosphere (and therefore is well-mixed). Together these imply that $CO_2$ mole fraction changes due to NEE over large regions (e.g., temperate North America) can be hard to distinguish from variability in $CO_2$ mole fraction resulting from flux processes and transport upwind. $CO_2$ mole fraction changes in $X_{CO2}$ from NEE at the surface are diluted over the path length of the atmosphere and largely obscured by meteorological variability (Basu et al., 2018; Feng et al., 2019).

*In-situ* measurements that comprise global networks, such as NOAA's Global Greenhouse Reference Network (https://www.esrl.noaa.gov/gmd/ccgg/ggrn.php), are rigorously carefully calibrated relative to the World Meteorological Organization (WMO) calibration scale (data used here are reported on the X2007 scale), thus ensuring the fidelity of these measurements over timescales of seasons to decades (Andrews et al., 2014; Hall et al., 2020). The open-path nature of space-based $X_{CO2}$ measurements does not allow for mole fraction calibration. Satellite retrievals require a forward model of radiative transfer that is run through an inversion system along with satellite-obtained absorption spectra of atmospheric $O_2$ and $CO_2$ to infer $X_{CO2}$. While a great amount of progress has been made to identify and eliminate sources of uncertainty emanating from this chain of processes, e.g., in the molecular absorption model and spectroscopy (Thompson et al., 2012; Payne et al., 2020; Hobbs et al., 2020), considerable sources of uncertainty remain. These are attributed to the presence of aerosols in the column (Connor et al., 2016), clouds and cloud shadows (Massie et al., 2021), interference of jointly retrieved parameters (Kulawik et al., 2019), surface properties, and details of the instrumentation. For example, Connor et al. (2016) estimate that aerosol dependent biases for retrievals over land may be as large as ~2 [ppm]. Moreover, sensors typically degrade over time, and limited information is available to characterize resulting time-dependent systematic errors. Post-launch data corrections are routinely performed and have generally reduced $X_{CO2}$ bias. For example, mean bias in land-nadir $X_{CO2}$ relative to TCCON in the *v8* product was reduced from 0.72 ± 1.22 [ppm] to 0.30 ± 1.04 [ppm] (O'Dell et al., 2018) and a correction in a geo-location error resulted in a decrease in *across-scene* standard deviation from 1.35 [ppm] in *v8* to 0.74 [ppm] in the *v9* data product (Kiel et al., 2019)."

*Currently, satellite derived XCO2 retrievals are linked to the WMO scale most directly through a limited set of in-situ profiles obtained over a network of ground-based Fourier Transform Infrared Spectrometers that comprise the Total Carbon Column Observation Network (TCCON; Wunch et al., 2017).*

Satellite retrievals of $X_{CO2}$ are linked to the WMO scale through thousands of coincident measurements over TCCON stations throughout the lifetime of the mission.

*However, TCCON itself provides remotely sensed information about XCO2, and TCCON retrievals undergo a complex validation and bias correction routine (Wunch et al., 2010) that*

*links these retrievals to the WMO scale.*

The TCCON data have been rigorously and systematically compared with in situ profiles that are, in turn, calibrated to the WMO X2007 scale. There have been about 80 such $CO_2$ in situ profiles over TCCON stations – the first collected in 2004, and the most recent from the current year. I do not know what the authors mean by "TCCON retrievals undergo a complex validation." The paper cited details the method of tying the TCCON retrievals to the WMO X2007 scale.

*Moreover, TCCON sites are few.*

There are 26 currently operating TCCON stations.

*Issues with validation of OCO-2 retrievals via TCCON have been identified- seasonal and site dependent biases have been reported (Wunch et al., 2017, 2015), raising questions about the adequacy of this network to validate satellite derived XCO2 products (Basu et al., 2013).*

I do not see how the 2015 paper (describing the GGG2014 algorithm and dataset) is related to this topic. The 2017 paper identifies biases in OCO-2, and not in the TCCON data – the 2013 paper discusses biases in the RemoTeC GOSAT retrievals relative to TCCON. I do not think these papers question the "adequacy" of the network to provide valuable validation information.

*This is especially important since systematic bias corrections of XCO2 from TCCON data are developed over small spatio-temporal scales and extrapolated globally (Wunch et al., 2011) for retrievals over land and ocean respectively (O'Dell et al., 2018).*

I do not understand what the authors are referring to in this statement. If the authors are referring to the systematic bias identification related to retrieval parameters like albedo, aerosol, etc., then I do not know what "small spatio-temporal scales" the authors are referring to. The 2011 work did not use TCCON data to identify these systematic biases – it used GOSAT data from the southern hemisphere south of 25S. Comparisons with the TCCON data showed that the bias correction improved the comparisons globally, and identified the scaling required to tie the bias corrected GOSAT $XCO_2$ to the WMO scale.

*OCO-2 retrievals are additionally corrected for bias by comparing with 4-D CO2 mole fraction fields from global inverse models, and a small-area approximation, but both methods are prone to smoothing across fine-scale variability in XCO2 (O'Dell et al., 2018; Corbin et al., 2008).*

I don't understand this statement at all. It should be *comforting* that bias corrections derived from these independent methods are consistent.

*While bias correction generally reduces inferred surface flux uncertainty when retrievals are assimilated in atmospheric inversions (Basu et al., 2013), even small retrieval errors can lead to large errors in inferred flux (Takagi et al., 2014; Chevallier et al., 2014).*

I agree.

*Biases in XCO2 from OCO-2, hereafter XretCO2 , have been identified, and found to be related to surface (e.g. pressure, albedo) and atmospheric (e.g. aerosol loading, sky condition) properties (Kiel et al., 2019).*

The Kiel et al., 2019 paper talks about a geolocation error and a meteorological reanalysis sampling error. I'm not sure why the authors attribute aerosol, albedo, cloud cover to this paper.

*However, systematic biases not accounted for in the v10 bias correction approach persist and therefore the measurement uncertainty associated with individual retrievals is believed to be at least twice the value currently reported in the OCO-2 data files (Eldering et al., 2017).*

I don't see how the authors can justify this claim with the citation provided. Version 10 was released in 2020, and the Eldering paper is dated in 2017.

Revised text # 2

"Currently, satellite derived $X_{CO_2}$ retrievals are linked to the WMO scale most directly through a set of in-situ aircraft and AirCore profiles obtained above a network consisting of 26 ground-based Fourier Transform Infrared Spectrometers that comprise the Total Carbon Column Observation Network (TCCON; Wunch et al., 2017). However, TCCON itself provides remotely sensed information about $XCO_2$, and comparison with aircraft profiles have revealed errors ~ 1 [ppm] (Wunch et al., 2011). Seasonal and site-dependent biases associated with validation of OCO-2 retrievals via TCCON have been reported (Wunch et al., 2017). OCO-2 retrievals are additionally corrected for bias by comparing with 4-D $CO_2$ mole fraction fields from global inverse models, and a small-area approximation, but both methods are prone to smoothing across fine-scale variability in $X_{CO_2}$ (ODell et al., 2018; Corbin et al., 2008). While bias correction generally reduces inferred surface flux uncertainty when retrievals are assimilated in atmospheric inversions (Basu et al., 2013), even small retrieval errors can lead to large errors in inferred flux (Takagi et al., 2014; Chevallier et al., 2014)."

*Thus, a dynamic method to routinely evaluate satellite retrievals is necessary.*

This statement does not follow from the previous two paragraphs. Define what is meant by "a dynamic method".

We have removed the word dynamic from this sentence.

Other comments:

Figure 7 and lines 360-364: There's a 4-ppm latitudinal gradient in the column across North America before and after optimizing biospheric fluxes. This is not "small" from a carbon cycle perspective and should be detectable from space.

The reviewer has pointed out here, as above, that the original manuscript was not clear enough in defining the intent of this work. That is, to understand the sensitivities of OCO-2 retrievals to surface fluxes and the consequent implications of small errors in those data. We have clarified the text in lines 349-356 of the revised text to reiterate that we are not assessing the

ability of OCO-2 to characterize large-scale gradients, but to identify smaller changes in NEE, for e.g., flux adjustments from biospheric flux models.

The revised text (lines 349-356) reads as follows:

"Considering that the vast majority of current inverse modeling or data assimilation systems used for $CO_2$ flux estimation are designed to correct errors in an *a priori* estimate, the effective flux signal is considerably smaller than shown above (right column in Fig. 4). In fact, when projected onto the total column, we find that the difference between a biospheric prior flux model (CASA-CMS) and flux optimized using *in-situ* observations over the continent by the CT-Lagrange inversion system (using the same prior flux estimate; Hu et al., 2019) for January and July, 2015 are $0.15 \pm 0.38$ and $0.09 \pm 1.06$ [ppm] respectively. Flux adjustment impacts on $X_{CO2}^{sim}$ are usually indistinguishable from 0 in January. While a gradient of $\sim 3$ [ppm] is visible across the continent in July (Fig. 7), less than a third of simulated retrievals show differences between prior and optimized flux greater than $\pm 1$ [ppm], when biospheric uptake over North America is strongest. Errors in terrestrial biosphere models of $CO_2$ flux translate to similarly small impacts on $X_{CO2}$, posing questions on the utility of these data currently in evaluating terrestrial biosphere models."

**References not already cited in the paper:**

Hooghiem, J. J. D., et al.: Wildfire smoke in the lower stratosphere identified by in situ CO observations, Atmos. Chem. Phys., 20, 13985–14003, https://doi.org/10.5194/acp-20-13985- 2020, 2020.

Hall, B. D., et al.: Revision of the World Meteorological Organization Global Atmosphere Watch (WMO/GAW) $CO_2$ calibration scale, Atmos. Meas. Tech., 14, 3015–3032, https://doi.org/10.5194/amt-14-3015-2021, 2021.

Crisp, D., et al.: The on-orbit performance of the Orbiting Carbon Observatory-2 (OCO-2) instrument and its radiometrically calibrated products, Atmos. Meas. Tech., 10, 59–81, https://doi.org/10.5194/amt-10-59-2017, 2017.

C. J. Bruegge et al.: Vicarious Calibration of Orbiting Carbon Observatory-2, in *IEEE Transactions on Geoscience and Remote Sensing*, vol. 57, no. 7, pp. 5135-5145, July 2019, doi: 10.1109/TGRS.2019.2897068.

Yu S, et al.: Stability Assessment of OCO-2 Radiometric Calibration Using Aqua MODIS as a Reference. *Remote Sensing*. 2020; 12(8):1269. https://doi.org/10.3390/rs12081269

New references cited

Hooghiem, J. J. D., et al.: Wildfire smoke in the lower stratosphere identified by in situ CO observations, Atmos. Chem. Phys., 20, 13985–14003, https://doi.org/10.5194/acp-20-13985- 2020, 2020.

Thompson, D. R., Chris Benner, D., Brown, L. R., Crisp, D., Malathy Devi, V., Jiang, Y., Natraj, V., Oyafuso, F., Sung, K., Wunch, D., Castaño, R., and Miller, C. E.: Atmospheric

validation of high accuracy CO 2 absorption coefficients for the OCO-2 mission, Journal of Quantitative Spectroscopy and Radiative Transfer, 113, 2265–2276, https://doi.org/10.1016/j.jqsrt.2012.05.021, 2012.

Payne, V. H., Drouin, B. J., Oyafuso, F., Kuai, L., Fisher, B. M., Sung, K., Nemchick, D., Crawford, T. J., Smyth, M., Crisp, D., Adkins, E., Hodges, J. T., Long, D. A., Mlawer, E. J., Merrelli, A., Lunny, E., and O'Dell, C. W.: Absorption coefficient (ABSCO) tables for the Orbiting Carbon Observatories: Version 5.1, Journal of Quantitative Spectroscopy and Radiative Transfer, 255, https://doi.org/10.1016/j.jqsrt.2020.107217, 2020.

Massie, S. T., Cronk, H., Merrelli, A., O'Dell, C., Sebastian Schmidt, K., Chen, H., and Baker, D.: Analysis of 3D cloud effects in OCO-2 XCO2 retrievals, Atmospheric Measurement Techniques, 14, 1475–1499, https://doi.org/10.5194/amt-14-1475-2021, 2021.

Hobbs, J. M., Drouin, B. J., Oyafuso, F., Payne, V. H., Gunson, M. R., McDuffie, J., and Mlawer, E. J.: Spectroscopic uncertainty impacts on OCO-2/3 retrievals of XCO2, Journal of Quantitative Spectroscopy and Radiative Transfer, 257, https://doi.org/10.1016/j.jqsrt.2020.107360, 2020.

Connor, B., Bösch, H., McDuffie, J., Taylor, T., Fu, D., Frankenberg, C., O'Dell, C., Payne, V. H., Gunson, M., Pollock, R., Hobbs, J., Oyafuso, F., and Jiang, Y.: Quantification of uncertainties in OCO-2 measurements of XCO2: Simulations and linear error analysis, Atmospheric Measurement Techniques, 9, 5227–5238, https://doi.org/10.5194/amt-9-5227-2016, 2016.

Wunch, D., Toon, G. C., Blavier, J. F. L., Washenfelder, R. A., Notholt, J., Connor, B. J., Griffith, D. W., Sherlock, V., and Wennberg, P. O.: The total carbon column observing network, Philosophical Transactions of the Royal Society A: Mathematical, Physical and Engineering Sciences, 369, 2087–2112, https://doi.org/10.1098/rsta.2010.0240, 2011